# Limited Differences in Insect Herbivory on Young White Spruce Growing in Small Open Plantations and under Natural Canopies in Boreal Mixed Forests

**DOI:** 10.3390/insects15030196

**Published:** 2024-03-15

**Authors:** Allison Pamela Yataco, Sabina Noor, Miguel Montoro Girona, Timothy Work, Emma Despland

**Affiliations:** 1Department of Biology, Concordia University, Montreal, QC H4B 1R2, Canada; pamela.yataco.marques@gmail.com (A.P.Y.); emma.despland@concordia.ca (E.D.); 2Ecological Research Group in MRC Abitibi (GREMA), Institute for Forest, Université du Québec en Abitibi-Témiscamingue, Amos, QC J9T 2L8, Canada; miguel.montoro@uqat.ca; 3Grupo de Análisis y Planificación del Medio Natural, Universidad de Huelva, Dr. Cantero Cuadrado 6, 21004 Huelva, Spain; 4Département des Sciences Biologiques, Université du Québec à Montréal, Montreal, QC H2X 1Y4, Canada; work.timothy@uqam.ca

**Keywords:** damage, galls, herbivory, insects, white spruce, plantation

## Abstract

**Simple Summary:**

We conducted this study in the boreal forest of Québec, Canada, to compare insect damage on white spruce (*Picea glauca*) trees within open-canopy, multispecies plantations, and trees naturally regenerated under mature mixed-wood canopies. Our plantation sites were established post-clearcutting and compared with naturally regenerated post-fire sites dominated by trembling aspen. Over two years, observations were made on young trees from ten sites in each environment, focusing on overall rates of herbivory, galls, spruce budworm damage, and sawfly defoliation. Despite a minor increase in spruce budworm damage in under-canopy forests from 2020 to 2021, overall, damage levels remained low, suggesting minimal impact on tree growth or mortality at this early outbreak stage. Contrary to initial assumptions, we found that insect damage was comparably low in both plantation and naturally regenerated environments, casting doubt on the idea that the enhanced sunlight and accelerated growth observed in plantations necessarily increase their susceptibility to pest attacks. In the context of global forest restoration efforts and Canada’s pledge to plant 2 billion trees by 2050, the resilience of white spruce plantations to insect damage is of paramount importance. Amidst concerns over increasing pest damage due to climate change, the observed resilience of multispecies plantations underpins the effectiveness of ecosystem-based management in maintaining low insect damage levels without compromising growth. This balance highlights the potential of mixed-species plantations to mimic natural forest conditions, contributing to biodiversity conservation and sustainable forestry practices. This research illuminates the intricate dynamics between tree growth, environmental conditions, and pest vulnerability, offering valuable insights for future forest management and conservation strategies.

**Abstract:**

In managed boreal forests, both plantations and natural regeneration are used to re-establish a cohort of conifer trees following harvest or disturbance. Young trees in open plantations generally grow more rapidly than under forest canopies, but more rapid growth could be compromised by greater insect damage. We compared insect damage on white spruce (*Picea glauca* (Moench) Voss, Pinaceae) growing in plantations with naturally regenerated trees under mature forest canopies in boreal forests (Québec, Canada). We selected ten sites in the naturally regenerated forest and in small, multispecies plantations and sampled ten young trees of 2.5–3 m (per site) in late summer 2020 and again in early and late summer 2021. We compared overall rates of herbivory, galls (adelgids), damage by the spruce budworm (*Choristoneura fumiferana*, Clemens), and defoliation from sawflies. Overall, insect herbivory damage remained at similarly low levels in both habitats; an average of 9.3% of expanding shoots were damaged on forest trees and 7.7% in plantation trees. Spruce budworm damage increased from 2020 to 2021 and remained higher in under-canopy trees, but damage rates were negligible at this early stage of the outbreak (1.5% in forest vs. 0.78% of buds damaged on plantation trees). While damage due to galls was higher in plantations, the overall low level of damage likely does not pose a significant impact on the growth or mortality of young trees.

## 1. Introduction

Forest plantations are widely used to produce wood fiber and re-establish forests after harvest. However, plantations can be particularly vulnerable to damage by herbivorous insects [1,2]. Two main mechanisms make plantation trees more vulnerable to insect pests. First, the open-canopy structure of plantations leads to higher temperatures that accelerate insect development and promote insect population growth. Increased solar radiation exposure in plantations alters leaf quality for herbivores, and sun-exposed leaves often have higher nutritional value and fewer defensive compounds [3,4] but can also be tougher [5]. Second, plant diversity and structural complexity are often much lower in plantations than in natural forests. It has long been understood that less complex ecosystems tend to exhibit far more extreme fluctuations in the population density of individual species [6,7] and are, hence, more vulnerable to outbreaks [8]. Indeed, higher plant diversity reduces defoliation by pest insects and reduces the risk of outbreaks [1,9,10,11,12]. Recent advancements in forest management strongly suggest that mixed-species plantations exhibit lower susceptibility to disturbances and are more beneficial for biodiversity and forest health compared to monocultures [13]. This study tests whether white spruce saplings in such high-diversity plantations are more vulnerable to insect damage than those regenerating under forest cover.

White spruce (*Picea glauca* (Moench) Voss, Pinaceae) is the main species used in reforestation in Canada [14]. White spruce is a shade-tolerant species that generally regenerates under mature stands of early successional deciduous trees; growth is faster in open sunlight in plantations, but trees in these open conditions are expected to harbor different insect communities [15] and have been observed to suffer higher levels of damage [16,17]. Native insect pest species on young white spruce have been well described [16,17], but with climate disruption, boreal forests are becoming more vulnerable to insect damage linked to invasive species and range expansion and shifts in the voltinism and phenology of native species [18].

The most significant insect pest on white spruce is the *Choristoneura fumiferana* (Clemens) (Lepidoptera, Tortricidae; commonly known as Eastern spruce budworm), whose outbreaks can last for decades. Over the past century in Québec (Canada), it has affected over 84 million hectares of forest [19]. Spruce budworm attacks mostly mature trees, but damage to saplings has been shown to be higher in open clearcuts than in partial cuts that retain some canopy cover [20].

Several other pest insects cause more damage in plantation saplings than in either mature trees or those subjected to under-canopy regeneration; however, mechanisms are complex and species-specific. For example, *Zeiraphera canadensis* (Mutuura & Freeman) (Lepidoptera, Tortricidae; spruce budmoth) show a preference to feed on open-habitat trees, attacking leaders and slowing vertical growth [21,22]. *Pikonema alaskensis* (Rohwer) (Hymenoptera, Tenthredinidae; yellow-headed sawfly) attacks young spruce in both plantations and naturally regenerated stands, but open-grown saplings are more vulnerable to defoliation [23,24] because females prefer to lay eggs in full sunlight, and this behavior is associated with the phenology of shoot development in spring. Both *Adelges abietis* (Linnaeus) and *Adelges cooleyi* (Gillette) (Hemiptera, Adelgidae; Eastern spruce gall adelgid and Cooley adelgid) warp shoots of young trees, especially under open plantation conditions [16]. *Pissodes strobi* (W.D. Peck) (Coleoptera, Curculionidae; white pine weevil) larvae feed on terminal leaders of young trees, and damage is much higher on trees in full sunlight [16].

Unlike traditional plantation forestry, which often focuses on the production of a few species, ecosystem-based management aims to mimic the forest as a whole system, including its biodiversity, water cycles, disturbance rates, and carbon storage capacities. This distinction is crucial as it underpins a more holistic management regime that aligns with sustainable forest management principles [25]. Plants experience context-dependent tradeoffs between allocating resources to growth vs. defense and often exhibit less defended phenotypes in high-growth environments [26]. Indeed, white spruce trees in open-condition plantations have been suggested to attain higher growth rates but suffer higher herbivory than saplings regenerating under aspen cover [27]. We tested whether this hypothesis holds in small, diverse plantations under ecosystem-based management in the absence of an insect outbreak.

We examined patterns of defoliation on young *Picea glauca* (Moench) Voss (Pineaceae; white spruce) trees in the boreal mixed woods in Western Québec. The extensively managed multispecies plantations (spruce–jack–pine–larch–aspen) studied are small patches in a boreal mixed-wood matrix and were established under Québec’s framework of ecosystem-based management, and they are managed to maintain natural processes [28]. We tested whether trees in plantations were more susceptible to insect damage under endemic conditions than naturally regenerated understory trees. We compared growth rates between plantation and understory trees and tested for evidence of a tradeoff between growth and defense. We also evaluated differences in temperature, light and water availability, and leaf toughness to examine their role in mediating insect damage [29]. We hypothesize that the ecosystem-based management approach recovers some of the processes linked to forest cover but that abiotic factors and plant traits linked to open conditions could still promote higher herbivory levels.

## 2. Materials and Methods

### 2.1. Study Sites and Plot Selection

We scored herbivory damage in the Forêt d’Enseignement et de Recherche du Lac Duparquet (FERLD) in Western Québec (45°34′19.848′′ N–79°22′7.644′′ W). We selected ten plantations and ten aspen stands with white spruce understory regeneration (Figure 1).

Plantation sites varied in size (ranging between 2.5 and 23.9 hectares) and were a mix of both softwoods and hardwoods with tree heights of 1.5 to 3.4 m tall. Following clearcutting in 2006–2010, plantation sites were prepared by shredding the fallen or wood logs to provide a seedbed for the development of seedlings. Plantations consisted of small, 100 m^2^ plots of white spruce planted at 2–3 m intervals, located in a matrix of stands of different conifers and broad-leaved trees. These plantations do not use fertilizers or pesticides, in contrast to common practices used for white spruce production. Brush cutting to remove competitive vegetation was only performed once, seven years after planting. The genetic sources for seeds were local trees.

The forest sites are naturally regenerated after fire and range between 30 and 90 years old. Post-fire regeneration on mesic sites in this region is dominated by trembling aspen (*Populus tremuloides* (Michx)) with gradual replacement by softwoods balsam fir (*Abies balsamea* ((L.) Mill.)) and white (*Picea glauca*) and black spruce (*Picea mariana* ((Mill.) Britton, Sterns & Poggenburg)) as succession progresses [30]. Our forest sites are dominated by mature trembling aspen (50–60%), with an understory of young balsam fir (~30%) and spruce (black and white; 10%) (for more details on the stand characteristics, see Appendix A, extracted from Foret Ouverte).

### 2.2. Sampling Techniques

In both plantation and forest habitats, ten 2–3.5 m tall white spruce trees, at least 5 m apart, were selected at each site (see detailed explanation in this section below). All sites were separated by 100 to 2500 m. The first sampling period was conducted in July–August 2020. In 2021, the same trees were sampled in May–June (referred to as early summer) and again from mid-July to August (referred to as late summer). We assessed the damage on current-year growth, examining buds as they opened and began to expand in early summer and examining the expanding shoots that developed out of those buds in late summer. Two different methods were used, branch sampling at the end of the growing season (2020 and 2021) and timed surveys on whole trees (early and late summer 2021).

### 2.3. Branch Sampling

In late summer 2020 and 2021, we cut a 40 cm branch section from the mid-crown of each tree and examined current-year growth for damage in the lab using a dissecting microscope within 24 h of sampling. Herbivory patterns were scored for bud loss by *Dasineura piceae* (Felt) (Diptera, Cecidomyiidae; spruce gall midge), formerly *Mayetiola piceae*, causing disfigured brooming galls; *Adelges abietis* and *Adelges cooleyi*, attributed to gall-makers; *Dasineura swainei* (Felt) (Diptera, Cecidomyiidae; spruce bud midge) attacks, causing terminal bud damage; shoot damage caused by moth larvae, including the spruce budworm and spruce bud moth; and sawfly damage (described by Wilson 1977). *Pikonema alaskensis* (Rohwer) and *Pikonema dimmockii* (Cresson) (Hymenoptera, *Tenthredinidae*; sawflies) produce a unique pattern, in which needles are stripped from one side of the developing shoot, progressing with the season such that, in the late season, shoots become completely bare of needles. We also observed a high rate of rusty buds’ injury, which we suspected to be caused by a fungal pathogen, but since it is not attributed to insects, it was not included in the analyses. A visual estimation of total defoliation using the Fettes method following [31] was obtained only for late summer 2021, on each current-year shoot per branch. This method involves visually estimating the proportion of needles missing on a shoot according to pre-established categories. Next, each defoliation level was replaced with the category midpoint, and these values were averaged to derive an overall defoliation level for each branch, which was used in statistical analysis [32]. We also counted the total number of developed and undeveloped buds on each branch; a developed bud gave rise to a current-year shoot, whereas an undeveloped one remained in the bud stage.

### 2.4. Whole-Tree Surveys

We conducted a 3 min timed survey for damages on individual trees in the field in both early and late summer 2021 to capture damage by insects with different phonologies. A trained observer scanned current-year growth on branches at ca 1.5 m above the ground for a 3 min interval, recording any damage according to the categories described above. Spruce budworm damage was considered a separate category, as it was possible to distinguish it from damage from other caterpillars in the field. However, the damage caused by spruce gall midge and spruce bud moth could not be clearly distinguished in the surveys due to time limitations and accessibility and thus was not included. The total number of developed buds (i.e., buds that had opened and begun to expand) on the scanned branches was also counted. This method is faster than branch sampling and allows for the examination of a larger number of expanding shoots but might be less precise at detecting small damages.

### 2.5. Environmental and Tree Variables

For each collection period, we measured soil temperature (°C), soil humidity (%), and canopy cover (%) in forest and plantation sites on the same day. Tree height (m), lateral growth rates for multiple years (2018–2021), and needle toughness (g) were measured only in late summer. The canopy cover was measured via a spherical densiometer (model C, manufactured by Forest Densiometers, Rapid City, SD, USA). Annual elongation from 2018 onwards was measured on a lateral shoot approximately 1.5 m high during late summer sampling. Needle toughness was assessed on current-year growth at the end of the season using a penetrometer following [5]. We measured 10 individual needles per tree and pooled them as a single value, and only 4 trees per site were used in each habitat.

### 2.6. Statistical Analysis

In 2021, one forest site became inaccessible due to bear traps and was excluded from data analysis. We compared environmental parameters and tree traits to confirm predictions about sun exposure and tree growth. We used *T*-tests and Mann–Whitney U tests, depending on the normality assumptions. Spearman’s correlation analysis was utilized to investigate a potential association between damage types and measured environmental variables.

We next compared the damage to current-year growth in both datasets (branch data for 2020 and 2021; survey data for early and late summer 2021). Since variance was not constant across our observations in our data, we used the “glmer.nb” function from lme4 v. 1.1-34 [33] for fitting mixed-effect models (with negative binomial distribution) and “testUniformity” and “testDispersion” from DHARMa v 0.4.6 [34] to account for under- and overdispersion in our mixed-effect models. To visualize the statistical models, ggplot2 v. 3.4.3, and the “grid_arrange” function from gridExtra v. 2.3 [35] were used. All the packages were deployed in RStudio (v. 4.3.1). We specified our negative binomial mixed-effect models with two main explanatory variables: (i) habitat (plantation vs. forest) and (ii) sampling period (year for branch data and season for survey data). Stand (site) was designated as a random effect to account for potential variance. A model was applied independently to each damage type. In cases where the model encountered singularity issues, the interaction term between habitat and year was removed to address collinearity. To enhance model convergence and stability, the “bobyqa” optimization algorithm was implemented using the “glmerControl” function (control = glmerControl(optimizer = “bobyqa”)). This algorithm adjustment was applied selectively to ensure reliable results across all analyses. A generalized linear model with negative binomial distribution was utilized with the “glm.nb” function from MASS v 7.3-60.0.1 [36] to investigate the defoliation (through the Fettes predetermined defoliation classes) in treatments on the 2021 foliage only. In our initial investigation, we incorporated potential environmental variables, identified through Spearman correlation analysis (lateral growth of current and years), into our mixed models. However, the results failed to yield statistically significant associations with individual damage types. Moreover, the inclusion of these variables did not lead to a distinct improvement in AIC scores (compared via ANOVA between simple and complex models). Consequently, these variables were not further integrated into our subsequent analyses, and we stuck to a simple model.

## 3. Results

The *t*-test results are consolidated in a Appendix A. As predicted, canopy cover (%) was greater in forests, with an average of 87%, while plantations had an average cover of 37% (Appendix A). Soil humidity and temperature were higher in plantations than in forest sites (Appendix A). Lateral shoot growth varied between years but was consistently significantly higher in plantation than in forest understory trees. The toughness of mature needles was slightly higher in young spruce trees in plantations (56.4 g) than in forest trees (50.2 g) (Appendix A). We looked at how different environmental variables might be connected to damage to trees using branch data. Spearman’s correlation showed that increased lateral shoot growth in 2021 tended to have more developed buds, galls, and spruce budworms. There was a connection between increased lateral shoot growth in 2020 and caterpillar damage as well as total buds being damaged. Also, with the increase in canopy cover, forest stands tended to have more damage from spruce budmoth. However, needle toughness, soil temperature, and soil moisture did not show significant relationships (Appendix A).

### 3.1. Branch Damage

The total number of buds that developed into shoots was similar in the two habitats (see Table 1). Damage from spruce gall midge and spruce bud midge was too infrequent to be included in the analysis. Data are summarized in Table 2 and display the outcomes of mixed-effect models, and graphical illustrations of the expected mean counts of damaged buds are plotted in Figure 2a–d. Total shoots damaged showed no significant difference between forest and plantation trees, but the number of damaged shoots increased significantly from 2020 to 2021. The number of galls was significantly higher on plantations than on understory trees. This damage increased in 2021 relative to 2020. Damage by all caterpillars, including the spruce budworm, did not differ significantly between forest and plantation trees; however, it did increase significantly from 2020 to 2021 in line with the progression of the spruce budworm outbreak in the region. Sawfly damage appeared slightly higher on understory trees, but not significantly so, and this activity showed no difference between the two years. Expressed as a proportion of expanding shoots affected, damage in late summer was recorded on average in 9% (CI: 0.050–0.136) of shoots in forest trees and 7% (CI: 0.039–0.115) of shoots in plantation trees.

Overall, defoliation (through the Fettes method) on current-year growth remained similar in both habitats for the observed classes. The results from the negative binomial generalized linear model revealed no significant defoliation difference in our habitats (see Figure 3). As such, both the forest and plantation had an overall defoliation of class 1 = 0–10%.

### 3.2. Timed Field Surveys

A statistical summary of damage patterns recorded via timed survey is summarized in Table 3. The outcomes of our mixed-effect negative binomial regression models for timed surveys are integrated in Table 4, and graphical illustrations of the expected mean counts of damaged buds are plotted in Figure 4a–e. The total number of damaged buds/shoots did not differ significantly between forest understory trees and plantation trees but did increase substantially over the growing season, suggesting the activity of insects with different phenologies. The prevalence of galls was higher in the plantation compared to the forest stands and increased between early and late summer, suggesting that many galls had not yet been formed or were too small to be detected during the first sampling period. Spruce budworm damage did not differ significantly between forest and plantation trees, nor did it increase over the growing season, confirming that spruce budworm activity was well underway during the first sampling period and that the larvae did not, as a general rule, move into and attack new shoots. Expressed as a proportion of developing buds affected, the spruce budworm damaged only 1% (C.I: −0.009–0.025) of buds on forest trees and 0.78% (C.I: −0.010–0.040) of buds in plantations (in early summer). Damage by other lepidopteran larvae did not differ either between the two habitats but did increase over the growing season, suggesting the activity of some species with later phenology than the spruce budworm. Sawfly damage was more pronounced in the forest than in the plantation trees. It also increased in late summer, suggesting a relatively late phenology for sawfly species active in these sites.

## 4. Discussion

The overall level of damage by insect herbivores did not differ significantly between plantation white spruce trees and those in under-canopy regeneration. However, various insect-feeding guilds showed different trends: Gall-forming adelgids were more abundant in plantations, and sawflies appeared slightly more abundant in the understory. The branch data collected at the end of the growing season showed similar levels of damage in the two habitats over the two years of sampling. The field survey data showed that spruce budworm damage was more detectable in early summer, suggesting that it was potentially underestimated by late-season branch sampling. As expected, the trees grown in plantations experienced greater canopy openness, higher soil temperature, and humidity, and exhibited tougher needles and faster shoot growth. However, none of these variables were significant predictors of insect damage. We did not observe a growth–defense tradeoff [26], suggesting that the higher growth observed in plantations does not necessarily incur a significant cost in terms of greater herbivore damage.

Our study distinguished between plantation and forest environments based on their origins: The former developed from mixed softwood and hardwood stands post-clearcutting from 2006 to 2010, with site preparation involving shredding residual wood logs to promote seedling growth. In contrast, the latter emerged through natural regeneration after fire, aging from 30 to 90 years. Initially, forest sites are led by trembling aspen, gradually transitioning to a conifer-dominated landscape, including species such as balsam fir and both white and black spruce [30]. Despite expectations of higher sun exposure in plantations leading to increased tree growth and decreased resistance to insects [26,27], no relationship was detected between any damage types and factors such as canopy openness (an indirect measure of solar radiation), temperature, or tree growth rate. Sun exposure can affect insect performance, both directly by accelerating development and indirectly via plant traits, which can lead to conflicting results. Accelerated development at higher temperatures is common in in boreal forestinsectss, where temperatures are often below those optimal for growth [37].

The sun-exposed foliage of spruce trees emits higher levels of terpenoids than shade foliage [38], but the evidence for the effects of these compounds on leaf-feeding insects is at best mixed [39]. Foliage toughness is a significant defense of conifers against insect herbivores; previous works show that this is higher in sun-grown white spruce foliage [5], and our results confirm tougher foliage in plantations than in under-canopy regeneration stands. However, other studies suggest that spruces upregulate defense genes in shade [40]. Thus, it is not clear whether sun-grown or shade-grown foliage is more palatable to herbivores, and the different patterns observed with adelgids and sawflies in the present study suggest that the effect could differ between herbivore species. For instance, previous work suggested that lower-crown (i.e., shade-grown) foliage [41] could favor spruce budworm performance.

Indeed, in our study, different herbivorous insect-feeding guilds responded differently to the conditions in plantations compared to under-canopy white spruce trees. Much of the damage observed in plantations was attributable to galls. Increased gall damage in plantations is very common, as the spruce gall adelgid (*Adelges abietis*) favors more open habitats [42]. However, gall damage generally does not affect the growth rate of the shoots [43]. The highest rates of gall formation that we observed in late summer 2021 remained under 2% of affected shoots, which is considered a trace level of damage in the government monitoring of damage to plantations [44] and does not imply any measurable negative impact on young spruce trees. 

The spruce bud midge was only observed at very low levels (<0.1% of affected buds) in our branch sampling and was undetectable in survey sampling. However, government data show that 25% of Québec white spruce plantations were affected by this insect in 2021, with damage rates up to 25% of new growth attacked [44]. Similarly, the white pine weevil (*Pissodes strobi*) was observed on 9% of trees in 2021 in government-monitored white spruce plantations [44], but not at all in our study.

The sawfly defoliation we observed fits the pattern shown by *Pikonema alaskensis* and *P. dimmockii*, both species that were collected at the study site. Yellow-headed sawflies can severely defoliate young spruce in both plantations [24] and under-canopy forest trees across Canada [17]. Sawflies feed preferentially on developing foliage, but at high densities, and then backfeed on old needles and can cause tree mortality after 3–4 years of continuous defoliation [23]. Data from plantations across Québec in 2022 show that sawfly damage was relatively rare (2 of the 100 monitored trees impacted) but that it reached 15% of new foliage attacked in plantations where it was observed [44]. The damage observed in our study was at a trace rate (0.8% of buds per stand), with damage slightly lower in plantations than in under-canopy trees (1.8% of buds). This finding suggests that although some Québec plantations are at risk of sawfly damage even under endemic conditions, this is not the case for the plantations we studied.

Damage by caterpillars could not be attributed to individual species, but the field survey suggested that it was not entirely due to the spruce budworm and that other species of later phenology were involved. Among others, the spruce budmoth has been known in the past to cause significant injuries in the plantations of Québec [45], but in 2021, it only caused trace (<5%) to low (5.1–25%) levels of defoliation in government-monitored plantations [44], similar to what was observed in our study.

The early-season survey conducted in 2021 confirmed spruce budworm feeding on both plantation and under-canopy trees, consistent with the progressing outbreak in the study region [44]. The majority of trees (67%) in government-monitored plantations were attacked by the spruce budworm in 2021, with damage rates ranging from trace (<5%) to high (>70% of new growth attacked). As observed for the pests described above, the plantations in our study are among the less affected ones (0.6%). Previous work suggests that young trees in open conditions may be more vulnerable to spruce budworm defoliation than those with at least partial canopy cover [20] and links this difference to the effects of canopy opening on egg abundance and larval dispersion. Ovipositing female spruce budworm moths lay their eggs on accessible branches with high sun exposure, and egg densities are expected to be higher in open plantations than in under-canopy environments. Early-instar larvae disperse between host trees by ballooning, and under-canopy regeneration is expected to be protected from ballooning larvae by over-canopy branches [46,47]. As a result, the authors of [20] recommend at least partial canopy retention in areas heavily affected by the spruce budworm. The small size and position within a forest matrix of our plantations could provide them with similar protection as observed in [20] for partial cuts.

The Fettes estimation, which summarizes defoliation on current-year shoots from many sources, showed an average needle loss in both habitats ranging in class 1 (0–10%) in 2021. Previous work shows that young trees in plantations can suffer over 50% defoliation during spruce budworm outbreaks [48] and that mortality can occur starting after 1 year of 80% defoliation [48]. Light (10–35%) to moderate (35–70%) levels of defoliation in regeneration slows both terminal and radial growth, and the resulting canopy opening can increase competition from forbs, shrubs, and hardwoods that further compromise regeneration success [20]. The mean defoliation rate in the plantations in our study remains well below this level, but the spruce budworm outbreak is progressing in the region, and it remains to be seen how these plantations will be affected.

Ecosystem-based management in the context of boreal forests refers to an adaptive approach that aims to manage the forest in a manner that sustains its natural processes, diversity, and productivity over the long term. This strategy focuses on the conservation and sustainable use of landscapes to address both ecological and human needs amidst climate change and other environmental challenges [25,26]. Previous studies show that, compared to coniferous monocultures, mixed plantations more strongly mimic natural vegetation and have reduced susceptibility to disturbances, including insect damage [49]. While large monoculture plantations often suffer high levels of damage [13], the small, extensively managed mixed-species plantations in our study show damage rates that in some cases are higher than those observed in understory regeneration stands but that are in the lower range of damage rates observed in government-monitored plantations [44]. These minor losses are not likely to affect the growth of plantation trees [44]. The relatively low levels of damage we observed could be linked to a higher abundance of natural enemies in these mixed-species planted forests that are embedded in a mature forest matrix [50]. Indeed, mixed-species plantations are known to offer resilience [51] through the increased diversity and abundance of natural enemies such as predators and parasitoids that constrain the extent of herbivorous insect damage [50,52].

Under the global forest restoration initiative and by the 2019 pledge, Canada plans to plant 2 billion trees by 2050 [51,53], and white spruce is one of the most planted species. Damage from forest insect pests is expected to increase with global change, especially in the boreal zone, due to higher temperatures, invasive species, and northward range expansions, and Québec white spruce plantations have been observed to suffer heavy insect damage [18]. In response to the challenges encountered with white spruce plantations, previous work has suggested that planting white spruce under 40–80-year-old aspen might offer better establishment conditions [54]. This under-canopy planting is thought to incur several advtableantages, including lower competition from forbs and shrubs, lower risk of frost damage, and less insect damage, which could outweigh the reduced growth [27]. Our results show that, under endemic conditions, open plantations under ecosystem-based management can attain the low level of insect damage observed in under-canopy trees without a reduction in growth. However, the situation might differ under outbreak conditions, when the vulnerability of trees in open environments could increase [20]. The similarity we observed in damage between plantations and understory regeneration stands implies two significant points: (1) These plantations, in the absence of a major outbreak, do not appear to suffer increased threats from any of these defoliating insects, and (2) these mixed-species compositions effectively mimic natural forests as required under ecosystem-based management. This emphasizes their potential to contribute to biodiversity conservation and sustainable forestry practices.

## Figures and Tables

**Figure 1 insects-15-00196-f001:**
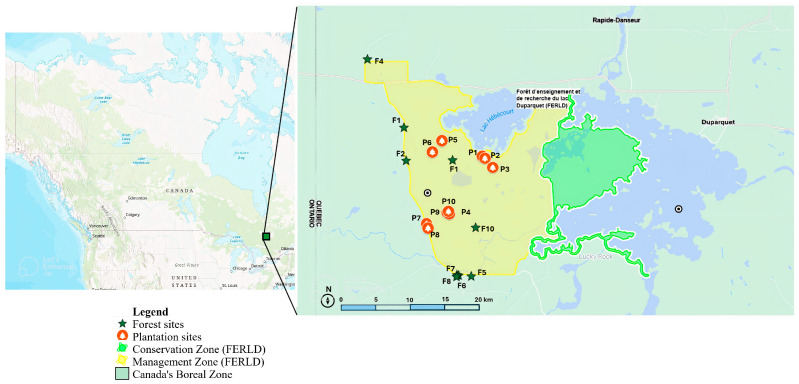
Map of Canada’s boreal zone with an enlarged section from the box, showing study sites in Abitibi Temiscamingue, Québec. The enlarged section displays the area around FERLD station where the two zones (management = yellow, and conservation = green) are highlighted in their respective colors. The selected habitat sites of plantations are indicated with tree icons (orange), and forest sites are shown in green stars. The scale bar displays the area cover of the entire FERLD-Duparquet region (including Lac Duparquet) with a north arrow embedded in the enlarged map. FERLD cartographic file was obtained from Forêt d’Enseignement et de Recherche du Lac Duparquet—Maps (uqat.ca), and mapping was executed via google maps and ArcGIS (online).

**Figure 2 insects-15-00196-f002:**
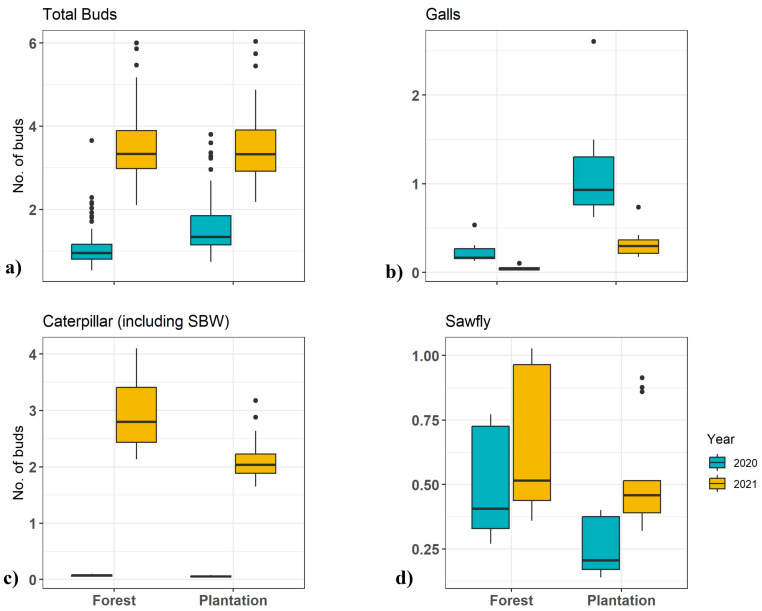
Boxplots of mean predicted counts of damaged white spruce buds by damage type and treatment over two years (2020–2021). The y-axis shows the number of buds affected, and the x-axis indicates the treatment type. The subplots detail (**a**) total buds being damaged from all insects; (**b**) gall damage; (**c**) caterpillar damage, including by spruce budworm; and (**d**) sawfly damage, comparing forest (under-canopy, naturally regenerated) and plantation settings. Results indicate a higher incidence of total and caterpillar damage in plantation environments, while gall and sawfly damage rates were less frequent.

**Figure 3 insects-15-00196-f003:**
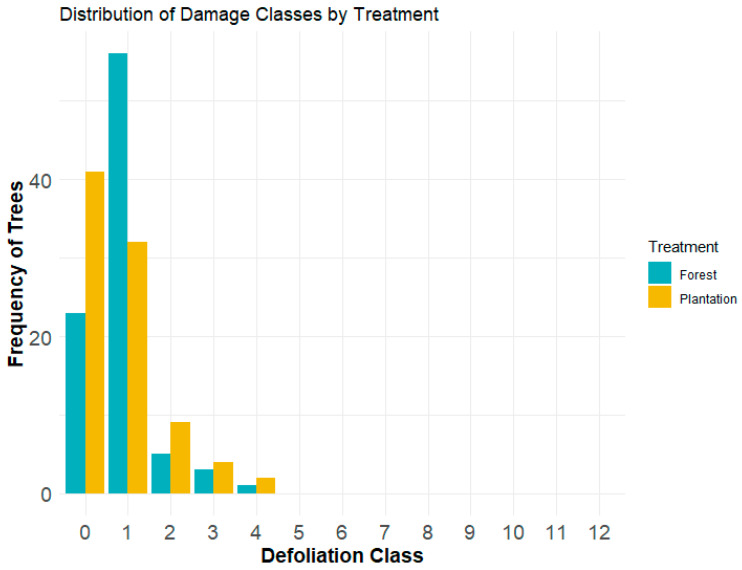
Grouped bar graph showing the frequency of white spruce trees across defoliation classes by treatment in forest and plantation habitats. Data represent 2021 foliage from a 40 cm branch segment, classified into 12 defoliation categories using the Fettes method (0 = 0%, 1 = 0–10%, 2 = 10–20%, …, 12 = 100+%). Each bar denotes the number of trees within a specific defoliation class for the two habitats. The graph highlights similar defoliation patterns between forest and plantation environments, primarily within the lower defoliation classes (0 and 1). Data were collected from ten trees across ten sites per treatment.

**Figure 4 insects-15-00196-f004:**
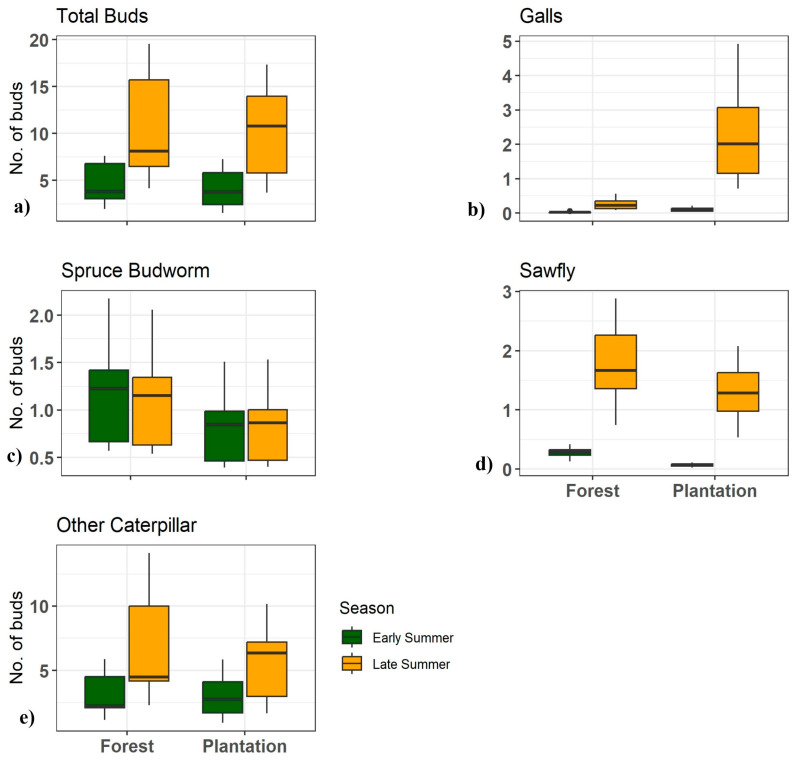
Boxplots illustrate the predicted mean counts of white spruce bud damage due to various insect-related injuries, across forest understory and plantation habitat. Damage categories are (**a**) total buds, (**b**) galls, (**c**) spruce budworm, (**d**) sawfly, and (**e**) lepidopteran larvae, with distinctions made between early- and late-season effects. The y-axis shows the number of damaged buds, and the x-axis categorizes the environments by treatment type. Legend colors and patterns highlight seasonal differences. Both habitats exhibit similar patterns of bud damage, with an increase observed across all categories during late summer. However, spruce budworm damage was comparatively higher in both seasons within the forest habitat.

**Table 1 insects-15-00196-t001:** Statistical summary of branch data for various damage types in two treatments of white spruce (2020–2021). Values represent the number of damaged shoots per sampled branch.

Treatment	Damage Type	Mean	SD	Median
Forest	Developed buds	24.581	12.315	22
	Undeveloped buds	1.785	2.506	1
	Fettes defoliation	0.447	0.626	0.211
	Galls	0.135	0.417	0
	Caterpillar (including SBW) damage	1.401	2.854	0
	Sawfly damage	0.593	1.383	0
	Spruce gall midge	0.0451	0.298	0
	Spruce bud midge	0.0734	0.337	0
	Total shoots damaged	2.293	3.21	1
Plantation	Developed buds	31.686	15.198	30
	Undeveloped buds	1.526	2.563	1
	Fettes defoliation	0.478	0.812	0.058
	Galls	0.792	1.593	0
	Caterpillar (including SBW) damage	1.186	2.473	0
	Sawfly damage	0.388	0.854	0
	Spruce bud moth	0.026	0.191	0
	Spruce gall midge	0.01	0.145	0
	Spruce bud midge	0.042	0.270	0
	Total shoots damaged	2.446	3.227	2

**Table 2 insects-15-00196-t002:** Mixed-effect negative binomial regression model results for damage types on white spruce stands analyzed for branch data (2020–2021). Fettes defoliation was assessed with a simple generalized linear model with a negative binomial distribution. A positive estimate paired with a significant *p*-value (less than 0.05) indicates a higher prevalence of the specified type of damage in plantation stands. On the other hand, a negative estimate points to a greater frequency of occurrence in naturally regenerated stands, as confirmed by the *p*-value’s significance.

	Estimate	Std. Error	z Value	Pr(>|z|)
Total buds damaged				
Habitat (Plantation)	0.079	0.198	0.398	0.690
Year (2021)	1.364	0.189	7.189	0.00
Interaction (Habitat:Year)	−0.121	0.255	−0.474	0.635
Gall damage				
Habitat (Plantation)	1.582	0.315	5.018	0.000
Year (2021)	−1.637	0.587	−2.787	0.005
Interaction (Habitat:Year)	0.372	0.651	0.570	0.568
Caterpillar (including SBW) damage				
Habitat (Plantation)	−0.255	0.193	−1.321	0.186
Year (2021)	3.712	0.318	11.663	0.00
Sawfly damage				
Habitat (Plantation)	−0.656	0.366	−1.790	0.073
Year (2021)	0.285	0.342	0.834	0.403
Interaction (Habitat:Year)	0.539	0.510	1.057	0.290
Defoliation by Fettes method				
Habitat (Plantation)	0.06705	0.22942	0.292	0.77

**Table 3 insects-15-00196-t003:** Statistical summary of survey data for various damage types in two treatments of white spruce (early and late summer 2021). Values represent the buds counted per tree during the timed field survey.

Treatment	Damage Type	Mean	SD	Median
Forest	Developed buds	109.214	71.492	113.5
	Galls	0.131	0.486	0
	Caterpillar damage	5.016	6.352	3
	Spruce budworm	1.186	1.885	0
	Sawfly damage	1.065	2.137	0
	Spruce bud midge	0.000	0.000	0
	Cooley adelgid	0.148	1.1634	0
	Total buds damaged	7.549	7.895	6
Plantation	Developed buds	138.162	64.163	136
	Galls	1.431	4.149	0
	Caterpillar damage	4.314	6.111	2
	Spruce budworm	0.791	1.074	0
	Sawfly damage	0.710	1.601	0
	Spruce bud midge	0.0253	0.235	0
	Cooley adelgid	0.005	0.071	0
	Total buds damaged	7.279	9.391	4

**Table 4 insects-15-00196-t004:** Mixed-effect negative binomial regression model results for damage types on white spruce stands analyzed for timed-survey data (early and late summer 2021). A positive estimate accompanied by a significant *p*-value (<0.05) suggests that the corresponding damage type was significantly more prevalent in plantation stands. Conversely, a negative estimate indicates a higher occurrence in natural regeneration stands, with significance determined by the *p*-value.

	Estimate	Std. Error	z Value	Pr(>|z|)
Total buds damaged				
Habitat (Plantation)	−0.233	0.147	−1.590	0.111
Season (Late Summer)	0.757	0.140	5.390	0.000
Interaction (Habitat: Season)	0.114	0.195	0.584	0.558
Gall damage				
Habitat (Plantation)	1.420	0.004	316.99	0.000
Season (Late Summer)	2.384	0.004	532.11	0.000
Interaction (Habitat: Season)	0.738	0.004	164.78	0.000
Spruce budworm damage				
Habitat (Plantation)	−0.365	0.223	−1.638	0.101
Season (Late Summer)	−0.056	0.211	−0.267	0.789
Interaction (Habitat: Season)	0.071	0.307	0.231	0.816
Other caterpillar damage				
Habitat (Plantation)	−0.197	0.185	−1.066	0.286
Season (Late Summer)	0.684	0.179	3.815	0.000
Interaction (Habitat: Season)	−0.130	0.249	−0.523	0.600
Sawfly damage				
Habitat (Plantation)	−1.483	0.460	−3.224	0.001
Season (Late Summer)	1.742	0.272	6.395	0.000
Interaction (Habitat: Season)	1.155	0.503	2.292	0.021

## Data Availability

Data generated and used in this work will be made available (currently in review) via https://doi.org/10.5061/dryad.bnzs7h4j4 (accessed on 1 March 2024).

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
