# Peer review of "Limited Differences in Insect Herbivory on Young White Spruce Growing in Small Open Plantations and under Natural Canopies in Boreal Mixed Forests"

_insects, 2024, doi:10.3390/insects15030196_

Round 1
Reviewer 1 Report
Comments and Suggestions for Authors
Review of “Limited differences in background insect herbivory on young white spruce growing in open plantations and under forest canopies” submission to Insects
The study is well-designed and the manuscript and the results and discussion well documented. There should be some broader perspective (in intro and discussion) comparing the small plantation plots assessed in this study, to larger near-monoculture tracts which are not considered. There are a large number of mostly minor revisions necessary to the format and graphics and some clarifications necessary to the methods and results. Authors need to pay more attention to detail about presentation of text and figures. These are detailed below by line number of Table/Figure number. Those with a star* require a substantive revision. Those without a star require less effort, but nevertheless are important to correct.
Line 1: Article not Articel
Title: Delete the word “background” in the title; it does not make sense and adds no meaning.
Lines 6,8,10: Add “Canada” after QC.
Line 9: do not break onto a new line, the footnote from its text!
L.13: “subsequent” – to what: clarify.
L.14: more not increased.
L. 19: clarify source of galls – adelgid and cecidomyiid?
Abstract: give approximate size (height and dbh) and age of trees assessed.
L.49: early-successional
L.57, 63, 65, and many others – At first mention of insect name, give scientific name, author (not date), Order: Family. At subsequent mentions give ONLY the common name. Same for plants.
L.58,81 and many others – decide either Québec or Quebec, always the same everywhere in ms.
L.82, give a quick summary of “multispecies” such as “aspen-fir-spruce mixture”
L.86, delete “also”
L.93, do not break heading from its text
L.96: 48°N, -76°W is not even close! Please correct this misinformation to reflect the true center (latitude, longitude) of your study area.
*Figure 1: both maps are of inferior quality and must be greatly improved. The provincial map needs to show the context of surrounding provinces and states as well as bodies of water; remember this journal has an international audience. The inset of the research forest is not nearly the correct shape, nor it is the shape of the “région administrative de l'Abitibi-Témiscamingue”. On the close-up map, the color key and legend is too small, the labels on the map too small, the field sites other than forest are not sufficiently legible, and bodies of water need to be shown – a good portion of the FERLD shown is actually a large lake!
Page 3: should state range of size (height, dbh) of trees sampled.
*Page 3: apparently the forest sites have a history of fire, and the plantation sites in contrast are post-clearcut? This should be clarified, and handled in the discussion as well.
Page 3 and elsewhere, strange spacing (e.g. after “damage” line 95, after “characteristics” line 116, and after “summer” line 125, is evident in places throughout. Please correct this.
L.120, below not ahead.
L.121, change to “sites were separated by 100 to 2500 meters.”
Page 4, see nomenclature comment for line 57 et al.
*L.140, “We also observed a high rate of rusty buds’ injury” – what is a high rate and why not include it in the analysis? Don’t be blind to factors other than insects.
*L.159, why was indistinguishable damage “not included”? It should be.
L.143, 171, pages 11, 12, etc.: cite references by number, not name and date, per journal format. L.148 delete “REF”.
L.166, needle toughness is defined in grams? Needs an explanation.
L. 168, add “SD” after “Rapid City”
L.177, start sentence with “We used”
L.217, caterpillar, not caterpillar’s
L219-220 is not a sentence. Start with “However,”
Table 1, Galls per branch, not simply “Galls” in Forest treatment damage type.
Table 2, do not split table across different pages.
Table 3, Galls per tree, not simply “Galls” in Forest treatment damage type.
Figure 2, explain box plots thoroughly, in the caption.
Line 261, “is summarized” not “has been assimilated”
*Figure 3 should show the distribution of damage classes in a grouped bar graph for each treatment. It is not correct to average percent defoliation classes. The graph must be modified to reflect the data as collected.
Figure 4, again (as in Fig.2), authors need to explain thoroughly in the caption, the box plots used, and related to that, why are there no outliers shown? Whereas the same colors (blue and yellow) are used for forest and plantation treatments, respectively, this figure should use a different pair of colors, to avoid inevitable confusion with Figure 4 categories of early summer and late summer.
L.321, i.e., not i-e
L.326, what do you mean by “growth rate of the tree”? – shoot elongation, wood accumulation … ?
L.333, cite again [40]
Refence format is not carefully presented. Titles should be capitalized in sentence form. Scientific names need to be italicized, and species names not capitalized – for example “Picea Glauca” should be “Picea glauca” and many other instances.
Comments on the Quality of English LanguageSeveral places need improvement as to expression and grammar, as noted in details.
Reviewer 2 Report
Comments and Suggestions for Authors
The main comments are reported in the file. Additionally, I suggest highlighting the ecological differences between the two habitats (e.g.: the number of possible predators or parasitoids of herbivorous insects present).

Reviewer 3 Report
Comments and Suggestions for Authors
Interesting results contrary to hypothesis, with different insect feeding guilds with some higher and some lower trends, but overall lower if not significant damage in the plantations. This is an important example that the higher growth does not always incur a tradeoff to insect feeding.
- line 22 in abstract has an extra space before 7.7
-some sentences have 2 spaces and some have 1, check carefully throughout
-beginning line 57, many references are included and written out so that they are not a part of the numbered system, or potentially they will be out of order sequentially. It looks like the reference numbering system needs to be edited and corrected.
-A background or more clear definition of "ecosystem-based management approach" is needed in the introduction. This phrase isn't used until the hypotheses are mentioned, so it would be rather confusing to readers who are not familiar with silvicultural techniques or the nuances of ecosystem-based management vs plantations (which is not explicitly defined). Later it sounds like the plantations are being described as ecosystem based management regimes. It is just not as clear as it could be the first time reading through. Adding mixed to the title may help some, such as "...insect herbivory on young white spruce in open, mixed plantations and under mixed forest canopies. "
-how big/how many hectares overall were the plantation sites that had plots set up in them? this could be more described in methods to have an idea of how much edge or large area this were associated with. The right side of the map in Figure 1 is fairly illegible to read the labels.
-line 187 needs a period.
-did you observe or count terminal leader damage for white pine weevil (if so, I did not see this articulated in methods), or where they monitored for just in the bud and branch sampling....which would miss them. This could be clarified since you did not report any in your narrative.
